# Exploring Patient Views and Acceptance of Multiparametric Magnetic Resonance Imaging for the Investigation of Suspected Prostate Cancer (the PACT Study): A Mixed-Methods Study Protocol

**DOI:** 10.3390/mps3020026

**Published:** 2020-03-28

**Authors:** Joseph M. Norris, Veeru Kasivisvanathan, Clare Allen, Rhys Ball, Alex Freeman, Maneesh Ghei, Alex Kirkham, Hayley C. Whitaker, Daniel Kelly, Mark Emberton

**Affiliations:** 1UCL Division of Surgery and Interventional Science, University College London, London W1W 7TY, UK; veeru.kasi@ucl.ac.uk (V.K.); hayley.whitaker@ucl.ac.uk (H.C.W.); m.emberton@ucl.ac.uk (M.E.); 2London Deanery of Urology, London, UK; 3Department of Urology, University College London Hospitals NHS Foundation Trust, London W1G 8PH, UK; 4Department of Urology, The Whittington Hospital, Whittington Health NHS Trust, London N19 5NF, UK; maneesh.ghei@nhs.net; 5Department of Radiology, University College London Hospitals NHS Foundation Trust, London W1G 8PH, UK; clare.allen1@nhs.net (C.A.); alexkirkham@nhs.net (A.K.); 6Department of Pathology, University College London Hospitals NHS Foundation Trust, London W1T 4EU, UK; rhys.ball@nhs.net (R.B.); alex.freeman2@nhs.net (A.F.); 7School of Healthcare Sciences, Cardiff University, Wales CF14 4XN, UK; kellydm@cardiff.ac.uk

**Keywords:** diagnostic pathway, multiparametric MRI, patient views, prostate cancer, risk stratification

## Abstract

BACKGROUND: The introduction of multiparametric magnetic resonance imaging (mpMRI) has improved the diagnosis of suspected prostate cancer, accurately risk-stratifying men before a biopsy. However, pre-biopsy mpMRI represents a significant deviation from the traditional approach of prostate specific antigen testing with subsequent systematic transrectal ultrasound-guided prostate biopsy and we have not yet explored the views of men who experience this new pathway. The purpose of the PACT study (PAtient views and aCceptance of mulTiparametric MRI) is to explore men’s perceptions of mpMRI. METHODS: PACT will be conducted at teaching hospitals in which mpMRI is central to the prostate cancer diagnostic pathway using a two-phase, mixed-methods, quantitative and qualitative approach. In phase I, men referred with suspected prostate cancer will complete detailed surveys to explore their views on the mpMRI-directed pathway compared to the traditional pathway and on what constitutes ‘significant’ prostate cancer. In phase II, these themes will be expanded upon with in-depth, semi-structured interviews. Qualitative data will be transcribed and thematically analysed, and quantitative questionnaire responses will be analysed statistically. DISCUSSION: PACT will provide the first detailed insight into patient perceptions on the use and acceptability of mpMRI. Furthermore, results from PACT will help contribute to the resolution of outstanding controversies that surround this technology.

## 1. Introduction

### 1.1. Multiparametric Prostate MRI

The introduction of multiparametric magnetic resonance imaging (mpMRI) has enhanced the risk stratification for men at risk of prostate cancer [1]. Precision imaging, delivered through mpMRI, has addressed longstanding drawbacks of the traditional approach to prostate cancer diagnosis and is now integrated into the 2019 National Institute for Health and Care Excellence (NICE) Guidelines for men with suspected prostate cancer [2,3]. Classically, men with suspected prostate cancer would undergo serum prostate specific antigen (PSA) testing in the community, followed by systematic (random) transrectal ultrasound-guided (TRUS) biopsies. This approach carries risks, including the over-detection of insignificant cancer, over-treatment of insignificant cancer and under-detection of significant cancer [4], largely because traditional TRUS-guided biopsy is blind to cancer location [5]. Moreover, combining PSA testing with systematic TRUS-guided biopsy has been shown to fail at identifying men at risk of premature prostate cancer-related death [6]. In contrast, mpMRI has excellent diagnostic accuracy in the detection of clinically significant prostate cancer and the use of mpMRI before prostate biopsy enables more accurate pre-biopsy risk-stratification and lesion identification [1]. Pre-biopsy triage with mpMRI has now demonstrated that a proportion of men could safely avoid TRUS-guided biopsy and its associated side-effects, including pain, bleeding, infection, sepsis and anxiety (which is distinct from the general distress of a cancer diagnosis) [7,8].

### 1.2. Patient Perceptions of Prostate mpMRI

The views of clinicians (primarily urologists, radiologists and oncologists) have generally been favourable toward mpMRI [9,10,11]. However, we have not yet explored the views of the men who experience this novel pathway, in depth. In one study, Ullrich and colleagues surveyed a mixed group of men (with and without prostate cancer) on their views on prostate mpMRI in Germany [12]. They found that the majority (68%) considered mpMRI to be a useful method to obtain a prostate cancer diagnosis. However, they also found that only a minority (29%) had personally experienced mpMRI and that few had any knowledge of the role that mpMRI might play in any new risk-stratification process. Whilst this work helps in understanding the views that men may have about mpMRI, it does not give enough detail to shape the way that this technology is both explained and delivered in practice [13,14]. A rigorous exploration of the perceptions that men have about the accuracy and utility of mpMRI would help to contribute to the resolution of many of the uncertainties and questions that still surround its use.

### 1.3. Uncertainties of Prostate mpMRI

A small proportion (10–20%) of significant prostate cancers go undetected by mpMRI [1]; however, the true sensitivity and specificity varies due to wide intra-reader and inter-reader variability. To date, it is unknown whether men with suspected prostate cancer are willing to balance the benefits and drawbacks of the new mpMRI-directed diagnostic pathway (in which men with non-suspicious mpMRI may forgo biopsy) as compared to the traditional systematic TRUS-guided biopsy approach.

The nature and acceptability (to clinicians and patients) of prostate cancer that is undetectable by mpMRI is fundamentally important due to the ramifications it has on how we manage negative pre-biopsy mpMRI in which no significant cancer is visible (mpMRI scores 1–2). This also affects prostate biopsy strategies in which we must decide whether to only biopsy visible mpMRI lesions, or whether the rest of the non-suspicious prostate should be sampled simultaneously. Eliciting and understanding the opinions that patients have on these important issues will help influence future clinical decision making.

Here is a summary of the key outstanding uncertainties and questions surrounding mpMRI:

• *Acceptance of the diagnostic accuracy of mpMRI*

Many clinicians consider a false negative rate of 10–20% for mpMRI to be excessive, despite traditional systematic TRUS-guided biopsy having a sensitivity of 48% for the detection of significant cancer [1]. At present, the level of patient acceptance of the diagnostic performance of these two approaches is unknown. If patients are willing to accept the level of risk of false negative mpMRI, this would provide support to the mpMRI-directed diagnostic pathway.

• *Agreement to forgo biopsy when the mpMRI is negative*

At present, centres that have embraced mpMRI often avoid a biopsy when there is no visible lesion (i.e., mpMRI scores 1–2). However, this approach is criticised by some, as it carries the risk of missing a small proportion of mpMRI-invisible significant prostate cancers. Establishing whether men agree with this mpMRI-directed strategy would help to justify the avoidance of biopsies in cases of non-suspicious pre-biopsy mpMRI.

• *Opinions on lesion-only targeting*

An mpMRI-targeted biopsy detects more clinically significant prostate cancer than a traditional systematic TRUS-guided biopsy [2]. However, many urologists still perform systematic biopsies in addition to targeted biopsies, based on concerns regarding mpMRI-invisible disease. By exploring patient perceptions regarding biopsy strategy, we may clarify whether men are supportive of lesion-only targeting or whether they desire to have their entire prostate sampled, despite the higher risks of the detection of insignificant disease and biopsy-related side-effects.

• *Perceptions of what constitutes ‘significant prostate cancer’*

The definition of clinically significant prostate cancer varies, but conventionally relies upon disease volume and pathological grade, as these are believed to be the strongest determinants of clinical outcome. However, the true definition of significance has yet to be established. It would be valuable to explore which aspects of prostate cancer are most significant to men, as their views would enrich this debate and help to inform diagnostic and treatment decisions, expanding beyond a purely histopathological definition.

The PACT study (PAtient views and aCceptance of mulTiparametric MRI) aims to explore men’s views on the role that mpMRI could play and its level of diagnostic acceptability, with a systematic two-phase, mixed quantitative and qualitative approach. The results from this study will constitute the first dedicated evidence to address this question.

## 2. Experimental Design

### 2.1. Background

The PACT study is a prospective, observational, multi-centre, mixed-methods cohort study that will include all patients referred with suspected prostate cancer. Patients will be recruited from prostate cancer assessment clinics that are embedded within the current diagnostic pathway (Figure 1). Using data from a pilot study at the same hospitals and using an appropriately powered sample size calculation, we estimate that approximately 122 patients will be needed for phase I (questionnaire study), which should take approximately six to eight months to recruit (Figure 2).

Mixed methods will be used to determine patient views on the identified topics surrounding prostate cancer and mpMRI (Figure 3). In phase I, patients will complete detailed surveys (adapted from previously validated questionnaires) containing both quantitative and qualitative questions on the accuracy and use of mpMRI. In phase II, a subset of patients will be recalled to undergo in-depth, semi-structured interviews to explore these topics in more detail.

To ensure that the study was reported to a high quality, the Standards for Reporting Qualitative Research (SRQR) was used to design the qualitative component of this study [15]. The SRQR consists of 21 items to improve the transparency of all aspects of qualitative research by providing clear standards for reporting qualitative research.

### 2.2. Research Question

How do men perceive the risks and benefits of mpMRI during the diagnostic experience for suspected prostate cancer?

### 2.3. Aims

Examine the views of men on the mpMRI diagnostic pathway, including its acceptability and associated risks, as compared to the traditional approach of random TRUS-guided biopsy.Explore the opinions of men on how clinically significant prostate cancer is currently defined.

### 2.4. Objectives

Ascertain the level of acceptability of the mpMRI pathway to men referred with suspected prostate cancer.Investigate the extent to which men with suspected prostate cancer are willing to tolerate the current false negative rate associated with mpMRI.Describe the factors that may affect whether patients are willing to accept the drawbacks of the modern mpMRI pathway, over those of the traditional systematic TRUS-guided biopsy approach.Explore men’s perceptions of different prostate biopsy strategies.Elucidate men’s views on what constitutes clinically significant prostate cancer, describing this from a patient perspective.Identify potential areas for the development of patient education and support materials, in light of these findings.

## 3. Procedure

### 3.1. Phase I: Questionnaire Study

#### 3.1.1. Inclusion Criteria

Adult male patients (over 18-years-old).Referred with suspected prostate cancer.Undergone prostate mpMRI.Informed consent given to complete questionnaire.

#### 3.1.2. Exclusion Criteria

Unable to read, write or understand English.Previous diagnosis of prostate cancer.

#### 3.1.3. Recruitment

All eligible patients who are referred to the prostate cancer assessment clinic, with suspected prostate cancer, will be invited to be included in the study, provided that they do not have a prior diagnosis of prostate cancer and have sufficient English language skills. All patients will be invited to be involved in the study, regardless of their MRI status. The purpose of the study will be explained to them, along with the proposed data collection methods. Any questions will be answered in detail. Patients who have had previous investigation (for example, a previous TRUS-biopsy) will not be excluded from the study, as this will enable these patients to compare their experience of the new diagnostic pathway with their experiences of the traditional approach. The target for recruitment in phase I is *n* = 122, as specified by our sample size calculation, based upon similar patient engagement research in prostate cancer [16]. The number of patients in each sub-group (for example, by age or ethnicity) will be determined by availability. The sample size was calculated using a standardised equation for the comparison of means:

*n* = [(Zα/2 + Zβ)(Zα/2 + Zβ) × (2 × (SD)(SD))]/(μ1 − μ2)(μ1 − μ2) [17]

Previous similar study data: Kazer et al. (2013) [16]

Where: μ1 − μ2 = 6.5; standard deviation (SD) = 12.8; Zα/2 = 1.96; Zβ = 0.84

*n* = [(1.96 + 0.84)(1.96 + 0.84) × (2 × (12.8 × 12.8))]/(6.5 × 6.5)

*n* = 61 (if study divided into two arms)

So, *n* = 122 (61 × 2)

#### 3.1.4. Data Collection

Once informed, willing patients will sign phase I consent forms (Appendix A) and be given copies of the patient information sheet (Appendix A) and study questionnaire (Appendix A). Having read and understood the accompanying information sheets, patients will complete and return questionnaires before leaving the clinic. The study questionnaire consists of a modified version of a validated questionnaire previously designed to elicit patient perceptions of cardiac MRI (permission from the original questionnaire authors has been sought and approved) [18]. Large fonts and clear English will be used to accommodate the full range of included patients, with varying eyesight levels, educational attainment and socioeconomic backgrounds.

### 3.2. Phase II: Interview Study

#### 3.2.1. Inclusion Criteria

Adult male patients (over 18-years-old).Referred with suspected prostate cancer.Undergone prostate mpMRI.Successful completion of the questionnaire in phase I, with informed consent given to be interviewed in phase II.

#### 3.2.2. Exclusion Criteria

Unable speak or understand English.Previous diagnosis of prostate cancer.No contact details given during questionnaire completion.

#### 3.2.3. Recruitment

All men included in phase I will be offered the opportunity to return to phase II to undergo semi-structured interviews to explore issues of relevance, in greater depth. To incentivise men to undergo study phase II, interviews will be conducted at the local hospital for each patient (to minimise disruption with travel and venue unfamiliarity) and patients will be fully informed about the importance of their involvement in helping to potentially shape the future of the diagnostic approaches for suspected prostate cancer. To further increase patient engagement, men will be encouraged to personally review their interview transcripts and any related research materials (for example, journal articles) that will be produced as a result of their involvement.

A target of 20 men will be recruited for the interview phase. This is based upon standard practice in qualitative research and will provide a balance of gaining a sufficient breadth and depth of responses [19]. In order to obtain a rich and diverse qualitative dataset, patients recruited to the interview phase will be drawn from varied ethnic, educational, socioeconomic and occupational backgrounds, and will have undergone a range of diagnostic experiences (for example, there will be a mixture of biopsy-naïve and biopsy-experienced patients) which will enable a more meaningful comparison of viewpoints.

#### 3.2.4. Data Collection

Interviews will be semi-structured using pre-determined topic guides (Table 1) and are expected to last between 20 and 30 minutes. After patients have had the interview process explained to them, and have been consented (Appendix A), the interviews will be recorded using an encrypted Dictaphone (Olympus WS-853) and performed in a quiet, pre-booked room on the respective hospital site. The semi-structured interviews will contain five different themes, as shown in Table 1. The questions will aim to explore men’s perceptions of prostate mpMRI, prostate biopsy techniques and definitions of clinically significant disease. They will be open-ended in nature, encouraging a conversational interview style, in which responses can be expanded upon whenever possible.

### 3.3. Data Analysis

#### 3.3.1. Quantitative Analysis

Both the questionnaire and interview phases will contribute to the exploration of patient views on prostate mpMRI. The questionnaire phase will precede the interview phase and findings from the questionnaire will be used to inform and evolve the focus of the interview process. Once the minimum number of patients have completed the quantitative aspects of the questionnaire study (the number required to meet our sample size; *n* = 122), informative statistical analysis will be possible. GraphPad Prism 8 (Graph-Pad Software, Inc., La Jolla, CA, USA) will be used for all statistical analyses of quantitative data and the generation of graphical representations of results. The level of statistical significance will be defined as *p* < 0.05.

Quantitative data from the outcome measures listed above will be analysed using the appropriate two-tailed statistical tests to assess for statistically significant differences in groups. All questionnaire responses will be converted to ordinal (Likert) values (e.g., 1–5), and the responses will be catalogued in an anonymised master database. Initially, all responses will be analysed with descriptive statistics. Ordinal values will be analysed with a *t*-test or the Mann-Whitney U test, depending upon the data distribution. When statistically appropriate, multivariate analysis will be undertaken. When possible, analysis will be stratified by key demographic characteristics of interest (for example, age, ethnicity, medical history and diagnostic experience) to assess for differences in these sub-groups.

#### 3.3.2. Qualitative Analysis

Due to the rich generation of data from the interview study, a smaller sample size (*n* = 20) will enable meaningful, qualitative analysis. Audio recordings of the semi-structured interviews will be transcribed verbatim by an independent scribe and NVivo 12 (QSR International Pty Ltd., Doncaster, South Yorkshire, UK) will be used for all analyses. Qualitative data will be analysed using thematic analysis which is “a method for systematically identifying, organising and offering insights into patterns of meaning (themes) across a dataset” [19].

This process consists of six key steps:**Step 1:** Transcripts will be read whilst re-listening to audio recordings to check accuracy and build dataset familiarity.**Step 2:** Transcripts will be coded to identify aspects of the data relevant to research objectives.**Step 3:** Codes will be collated into themes.**Step 4:** Themes will be reviewed with the broader research team to ensure code consistency within themes and to avoid overlap between themes.**Step 5:** A thematic map will be developed by refining themes and analysing their relationships.**Step 6:** The qualitative results will be collated and published using the words of participants to illustrate areas of agreement, as well as divergences of views.

## 4. Discussion

In this paper, we describe the methodological design of the PACT study and how quantitative and qualitative data will be synthesised in a two-phase, mixed-methods approach. To our knowledge, the PACT study is the first detailed mixed-methods project to explore patient perceptions of the use of mpMRI for the diagnosis of prostate cancer.

This mixed-methods study centres on the quantitative and qualitative assessment of men’s views on the benefits and drawbacks of the new diagnostic pathway for prostate cancer as directed by mpMRI, compared to the traditional systematic TRUS-guided biopsy pathway. It explores the perceptions that men have regarding the differing approaches to targeted prostate biopsies and their views on what the definition of truly significant prostate cancer is, as these are all key controversial areas in the field of prostate cancer diagnostics.

The findings of the PACT study will help clinicians and researchers to understand the views and potential concerns that men may have around the recent changes to the diagnostic pathway for suspected prostate cancer. This information will provide valuable patient-centred opinions on some of the uncertainty and controversial areas that still surround prostate mpMRI. From this, we can then direct future developments in prostate cancer diagnostic research and the generation of patient information and education materials, focused on the key areas highlighted in this study.

## Figures and Tables

**Figure 1 mps-03-00026-f001:**
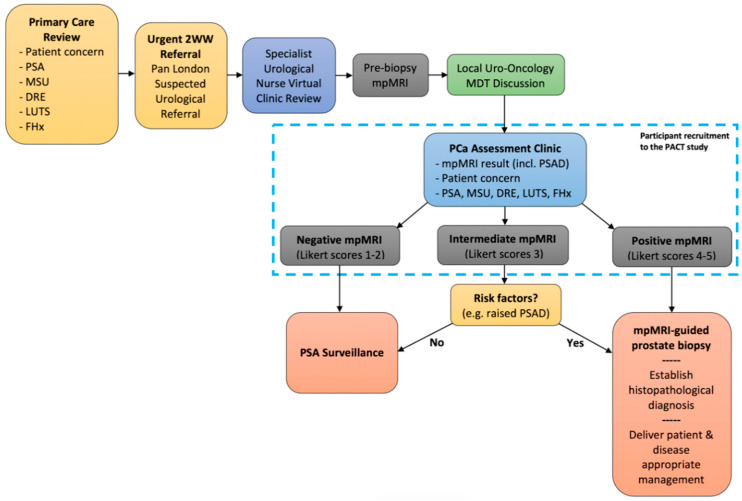
A diagram of the diagnostic pathway for suspected prostate cancer. The time interval between primary care referral and review in the prostate assessment clinic is less than two weeks. The time interval between mpMRI and MRI-guided biopsy is less than two weeks. *2WW* two week wait, *DRE* digital rectal examination (of the prostate), *FHx* family history (of prostate cancer), *LUTS* lower urinary tract symptoms, *MDT* multi-disciplinary team, *mpMRI* multiparametric magnetic resonance imaging, *MSU* mid-stream urine culture, *PCa* prostate cancer, *PSA* prostate specific antigen, *PSAD* PSA density.

**Figure 2 mps-03-00026-f002:**
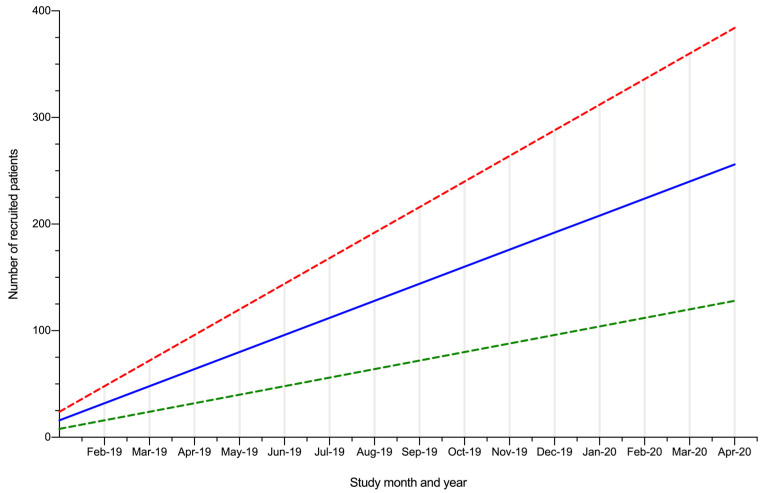
Predicted patient accruement (solid blue line: the expected rate of accruement, based on four eligible men being recruited per week; dashed green line: slower recruitment rate, based on two eligible men being recruited per week; dashed red line: accelerated recruitment rate, based on six eligible men being recruited per week).

**Figure 3 mps-03-00026-f003:**
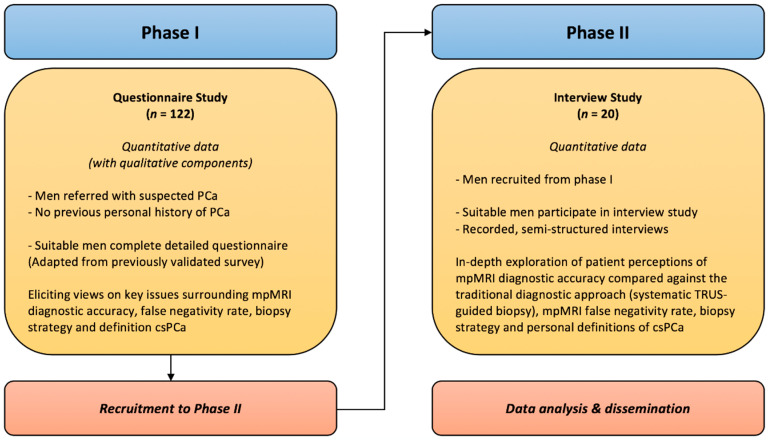
A flow diagram representing the different stages of the mixed-methods evaluation of the PACT study. *csPCa* clinically significant prostate cancer, *n* participant number, *mpMRI* multiparametric magnetic resonance imaging, *TRUS* transrectal ultrasound.

**Table 1 mps-03-00026-t001:** Themes (topic schedule) to be explored during the interview study.

Background	Pathway Comparison	Biopsy Strategy	Cancer Significance	Education
We have begun using MRI scans to help diagnose prostate cancer and would value your thoughts on this.	Previously, men with suspected prostate cancer would normally have prostate samples taken through the back-passage, without an MRI being performed – this approach could miss up to 50% of important cancers. Today, MRI scans are more readily available and can help us to detect around 80–90% of important tumours (if the MRI is performed before samples are taken).	When we take a prostate sample, we can choose to sample only cancers that we “see” on the MRI scan. This targeted approach has the advantage of diagnosing more “significant cancers” and fewer “insignificant” cancers; however, this does risk missing cancers elsewhere in the prostate. Alternatively, we can choose to sample the “whole prostate” (in addition to sampling the ‘target’) but this has a higher risk of detecting non-harmful cancer, bleeding, retention of urine and discomfort.	There is still no consensus definition of what makes prostate cancer ‘significant.’ However, some cancers are “more aggressive” than others and can spread around the body – these generally benefit from treatment. Others are “less aggressive” and do not spread or cause harm – these generally do not benefit from treatment and treating these cancers may cause more harm than the cancers themselves.	There are lots of information sources for patients and we would value your thoughts on these.
Can you tell me what experience you have had of MRI scanning? What do you understand about the use of MRI scans to detect prostate cancer? Are there particular aspects of MRI scanning that appeal to you? Are there any aspects of MRI scanning that you might have a concern about? Would you like to clarify these issues before proceeding?	Any new approach has drawbacks and benefits and the new MRI approach can miss 10–20% of prostate cancers – what do you think of this? How far does this risk worry you? With this in mind, how do you feel about not taking tissue samples at all when the MRI scan appears to be “normal” or shows “no cancer”? Can you tell me how you feel about this compared to the traditional approach? When comparing these two approaches (using MRI or not) which do you think is more appealing? And, why? From your point of view, are there any aspects of either of these approaches that would especially concern you? Given your own experience to date, can you think of any changes or improvements to these two pathway options that you would like to see being made?	Can you tell me, what experience you have had, if any, of prostate biopsy? Do you have any thoughts on which of the two major types of biopsy do you think that you would prefer? And, why is this? If you had an MRI scan that showed possible prostate cancer, would you wish to have: - Only a “target biopsy” (in which around 3–5 samples taken from the suspicious area shown on the MRI scan)? - Or, would you rather have a “targeted and systematic biopsy” (in which approximately 12 other samples are also taken from the rest of the prostate at the same time)? And why would you feel this way? As a related question, what proportion of important cancers would you be willing to miss?	If you were diagnosed with prostate cancer, which aspect of the cancer would be most important to you? Would you be more concerned about the effect on your quality of life (e.g., developing unpleasant symptoms, such as problems with urination) or your life expectancy (how long you might live for)? And, why might you feel this way? If you had prostate cancer, do you think you would you like to know about the potential danger posed by your cancer? In what way could we best provide such information to you? Do you think you would want to know about every single cancer that is present in your prostate (even “less aggressive” cancers) or do you think you would only want to know only about “more aggressive” cancers? Why would you feel this way?	What format is normally best for you when it comes to information about your health? Are there any aspects of this topic that you would like to have more information on? Can you summarise any key lessons for us about your diagnostic experience to date?
*MRI* magnetic resonance imaging.				

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
