# Peer review of "Exploring Patient Views and Acceptance of Multiparametric Magnetic Resonance Imaging for the Investigation of Suspected Prostate Cancer (the PACT Study): A Mixed-Methods Study Protocol"

_mps, 2020, doi:10.3390/mps3020026_

Round 1
Reviewer 1 Report
- Interesting study and well planned. Just a couple of questions that the authors should consider.
- Consider question to add in questionnaire
- If MR suspicious, do patients wish to have:
- Only target biopsy (~ 3 cores) , OR
- targeted + systematic biopsy (~ 3 + 12 = 15 cores)
- with higher chance of diagnosis of important prostate cancer but risk of higher morbidity (bleeding, retention of urine, discomfort)
- What is the proportion of important prostate cancer that you are willing to miss? (choose one) – [ This is an important quantitative question as the NPV of MRI for clinically significant
- 0%
- 5%
- 10%
- 20%
- 30%
- If MR suspicious, do patients wish to have:
- The age of the patient being recruited may affect the result. Would you stratify age groups?
- Older patients, say > 75, may accept a higher proportion of missed significant cancers.
Reviewer 2 Report
The study aims to explore men’s perceptions of MRI with a systematic two-phase, mixed quantitative, and qualitative approach. My specific comments are following.
- It was not clear how and why the exploration of the perceptions that men have about mpMRI would help the resolution of the uncertainties and questions (first, clarify the existing uncertainties and questions).
- More importantly, how would this survey study provide different information from the previous one by Ulrich et al.? Are we assuming that patients with suspected prostate cancer would have different perceptions about the accuracy and utility of mpMRI?
- Page 2, Line 69: Although it is well known that mpMRI has a high sensitivity for prostate cancer detection, the actual sensitivity and specificity of the clinically significant prostate cancer detection are highly dependent on the interpretation of mpMRI, which is known to contain substantial intra- and inter-reader variability.
- Page 2, Line 70-71: I do not understand what is supposed to be “traded” in the sentence. If the patients decided not to have mpMRI, they will likely suffer from over-treatment of insignificant cancer and under-detection of clinically significant cancer.
- Figure 1: The diagram is somewhat confusing. Why the patients go through two different mpMRI exams? Are the patients participated either before or after mpMRI? What is the expected time interval between mpMRI and MRI-guided biopsy?
- Page 6, Line 182: Please clarify how to incentivize the subject who returns to phase II.
- Table 1: The questions here are quite technical, and the authors assumed the participants had extensive knowledge of mpMRI and biopsy strategies, which would be less likely for most subjects.
Round 2
Reviewer 2 Report
Thanks for addressing my comments. Please see below for the remaining questions.
- Pathway comparison. It was not still clear what do we compare. For example, there would be three pathways, including MRI-directed (no biopsies for MRI negatives), no MRI (systematic biopsies), and hybrid (systematic biopsies for MRI negatives). If this is not clarified, the following questions may not be properly addressed.
- Does the study include subjects with MRI-negative? If not, how does the survey justify potential bias in response to balance the benefits and drawbacks?
- Biopsy strategy. Another benefit of using MRI-targeted biopsy is the reduced diagnosis of insignificant prostate cancer.
- Cancer significance. Most of the questions were confusing and not adequately addressing the current issue; no true definition of clinically significant prostate cancer has been established.
- The authors still assumed that the participants had significant knowledge of MRI and biopsy strategies.
Author Response
Please see the attached response document.

Round 3
Reviewer 2 Report
Thanks for addressing my previous comments. I have one remaining comment on Figure 1. It was not clear how MRI-negative patients would be recruited since the arrows are indicating MRI-guided prostate biopsy. Please revise it so that readers can easily recognize MRI-positive, MRI-negative, and MRI-indeterminate patients in recruitment.
Author Response
Please see attached response document.
